# Enhancing the diagnosis of CVD and depression comorbidity through the augmentation of synthetic metabolomics data

Vasileios C. Pezoulas,
*Unit of Medical Technology and Intelligent Information Systems, Dept. of Materials Science and Engineering, and BRI-FORTH, University of Ioannina, Ioannina, Greece,*
e-mail: v.pezoulas@uoi.gr

Nikolaos S. Tachos,
*Unit of Medical Technology and Intelligent Information Systems, Dept. of Materials Science and Engineering, and BRI-FORTH, University of Ioannina, Ioannina, Greece,*
e-mail: ntachos@bri.forth.gr

Georg Ehret,
*Department of Internal Medicine Hôpitaux Universitaires de Genève Geneva, Switzerland,*
e-mail: georg.ehret@hcuge.ch

Kevin Dobretz,
*Department of Internal Medicine Hôpitaux Universitaires de Genève Geneva, Switzerland,*
e-mail: Kevin.Dobretz@unige.ch

Dimitrios I. Fotiadis,
*Unit of Medical Technology and Intelligent Information Systems, Dept. of Materials Science and Engineering, and BRI-FORTH, University of Ioannina, Ioannina, Greece,*
e-mail: fotiadis@uoi.gr

Antonis I. Sakellarios*,
*Dept. of Mechanical and Aeronautics Engineering, University of Patras, Patras, Greece, and Unit of Medical Technology and Intelligent Information Systems, Dept. of Materials Science and Engineering, University of Ioannina, Ioannina, Greece,*
e-mail: asakellarios@upatras.gr

*Abstract*— **The cardiovascular disease (CVD) and depression comorbidity remains diagnostically challenging due to complex phenotypes and severe class imbalance in real-world health data. However, conventional resampling methods often fail to preserve the multimodal structure of high-dimensional clinical and metabolomics features. This study presents an AI-based pipeline applied to clinical and NMR-based metabolomics data from the UK Biobank to compare random downsampling against synthetic data augmentation using generative models like the conditional tabular generative adversarial network (CTGAN), the tabular variational autoencoder (TVAE), and the Tabular Denoising Diffusion Probabilistic Model (TabDDPM). The synthetic data produced by the CTGAN achieved the highest fidelity (Jensen-Shannon divergence 0.06 and average correlation difference 0.11 for the CVD diagnosis outcome). The AI models trained on synthetic data achieved superior performance across both classification tasks. For CVD diagnosis, the XGBoost reached 0.91 accuracy and 0.96 AUC, while for comorbid CVD and depression, 0.87 accuracy and 0.92 AUC. These results support synthetic augmentation as a robust solution to improve diagnostic performance across imbalanced datasets in healthcare.**

*Keywords*— *CVD, depression, metabolomics, synthetic data augmentation, AI.*

## I. INTRODUCTION

Cardiovascular disease (CVD) and depression are among the leading causes of global morbidity and mortality, frequently co-occurring in patients and jointly contributing to worse clinical outcomes. Artificial intelligence (AI)-based models have been increasingly adopted in this context, demonstrating promising results in early diagnosis and risk stratification. For example, gradient boosting and deep learning architectures have been used to predict major adverse cardiovascular events by incorporating both physiological and psychosocial data [1]. Explainable ML models tailored for comorbid conditions—such as coronary heart disease combined with depressive symptoms—have also been shown to support personalized care by highlighting actionable clinical features [2]. Furthermore, large-scale predictive modeling for depressive disorders using structured EHR data and unstructured clinical notes has demonstrated the utility of ML in improving diagnostic precision across mental health applications [3].

Despite these advancements, a persistent challenge in AI is the presence of imbalanced datasets. The clinical data often exhibit skewed distributions, particularly when modeling comorbidities, which can lead to biased classifiers with reduced sensitivity for underrepresented classes. To mitigate this, recent studies have explored synthetic data generation as a strategy to improve model training [4]. Traditional oversampling techniques, such as SMOTE or random duplication as well as random downsampling ones often fail to capture the complex, multimodal relationships present in high-dimensional tabular data. Emerging methods, including generative adversarial networks (GANs), variational autoencoders (VAEs), and diffusion-based generative models which have shown success in producing realistic synthetic health records that preserve statistical fidelity [5]. In addition to addressing class imbalance, synthetic data have been highlighted as a privacy-preserving alternative for data sharing and external validation [6].

In this study, we present an AI-driven pipeline for the joint diagnosis of CVD and depression using integrated clinical and NMR-based metabolomics data. The core technical contribution lies in systematically addressing class imbalance, a common limitation in comorbidity modeling. To evaluate the fidelity of synthetic data generation process, we conducted a comparative analysis to assess the ability of multiple synthetic data generators to replicate real data distributions including the conditional tabular generative adversarial network (CTGAN), the tabular variational autoencoder (TVAE), and a lightweight

implementation of the Tabular Denoising Diffusion Probabilistic Model (TabDDPM). Then we compared the conventional majority class downsampling strategy with synthetic data augmentation of the minority class using the synthetic data with the highest fidelity from the generators to enable targeted oversampling of the minority class while preserving the underlying data distribution. Both strategies are evaluated using multiple classification metrics across two predictive tasks. Our results delineate the conditions under which synthetic data augmentation outperforms conventional resampling methods (e.g., downsampling), particularly in AI modeling scenarios with limited minority class representation. The current work provides practical insights into the design and development of robust and generalizable AI-based models for diagnosis across imbalanced, multimodal medical data.

## II. MATERIALS AND METHODS

### A. The proposed workflow

Fig. 1 depicts the workflow of the end-to-end AI-driven approach which consists of four stages, namely: (i) the data engineering stage, (ii) the class imbalance handling stage, (iii) the AI modeling stage, and (iv) the AI model performance evaluation stage. More specifically, the data engineering stage involves: (i) the identification and removal of constant-value features that provide little to no discriminative power to eliminate noise, (ii) a stratified split of the noise-free dataset into training and testing subsets to preserve class distribution for fair evaluation, (iii) an optimized feature selection process using mutual information, where the selection is refined by detecting the optimal number of features via a knee/elbow point strategy.

The class imbalance handling stage aims to address class imbalance by utilizing generative-based synthetic data generators (e.g. the Conditional Tabular Generative Adversarial Network (CTGAN)) to enhance the representation of the minority class without duplication for better AI model generalization. In the AI modeling stage, the XGBoost algorithm is trained on both the resampled and the downsampled data. Finally, the AI model's performance is evaluated on the test set by calculating the accuracy, sensitivity, specificity, AUC, and F1 score. The output of the proposed workflow also includes ROC curves and feature importance plots from XGBoost's internal gain-based evaluation to provide deeper insights into the contribution of the selected features.

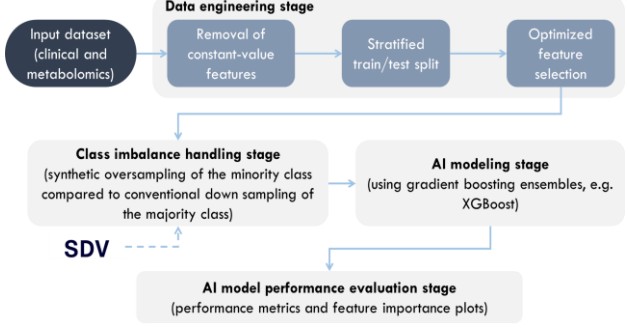

**Fig. 1.** The proposed workflow.

### B. Input dataset

The dataset used in this study integrates clinical information and nuclear magnetic resonance (NMR)-based metabolomics profiles, comprising 502,319 patient records and 318 features. In this study, we utilize data from the UK Biobank, a large-scale prospective cohort comprising over 500,000 participants, with extensive phenotypic, clinical, and omics data linked to longitudinal health outcomes [7]. The dataset provides a unique opportunity for integrative modeling of complex conditions such as cardiovascular disease and depression. For the current analysis, we curated a subset of participants with complete information across several domains: clinical biomarkers (e.g., blood pressure, lipid profiles), established CVD risk factors (e.g., BMI, smoking status, diabetes), biochemical assays (e.g., liver and renal function markers), social and behavioral variables (e.g., physical activity, alcohol consumption, socioeconomic status), and high-throughput metabolomics data obtained via nuclear magnetic resonance (NMR) spectroscopy [8]. This multi-modal dataset allows for the examination of both physiological and psychosocial determinants of health to enable the development of AI models which are sensitive to the multi-factorial nature of the CVD and depression comorbidity.

### C. Data engineering stage

Constant-value features were initially removed, including those that exhibit near-zero variance across all samples and can introduce redundancy and noise. This step was conducted to ensure that the input feature space retains only potentially informative variables. Stratified train/test split was then performed to partition the noise-free dataset while preserving the original distribution of the target classes across both subsets. In addition, this stratified process mitigates biases during the AI model evaluation since it ensures that minority and majority classes are proportionally represented in both the training and test sets. Feature selection was applied exclusively on the training subset to prevent data leakage. Mutual information (MI) was explored as a non-linear measure of dependency between features and the target variable and was further computed for all candidate features [9]. The knee/elbow point detection algorithm was finally used to identify the optimal number of features to retain.

### D. Class imbalance handling stage

Class imbalance is a well-documented issue in medical data which can severely impair the performance of supervised learning algorithms, particularly in identifying minority class instances. To address the inherent, skewed distribution of class labels, we explored a synthetic data-driven strategy for the generation of minority class samples using generative-based models like the Conditional Tabular Generative Adversarial Network (CTGAN). Specifically, the CTGAN from the synthetic data vault (SDV) library [10, 11] was utilized considering its ability to learn conditional distributions of features given class labels. In brief, CTGAN employs a mode-specific normalization strategy and training-by-sampling techniques to ensure stability and fidelity in the generated samples. It learns the conditional distribution $P(x|y)$ of features $x \in R^d$ given class label $y \in \{0,1\}$, through an adversarial network:

$$min_G max_D E_{x,y \sim P_{real}}[log D(x,y)] + E_{\tilde{x} \sim G(z,y)}[log(1 - D(\tilde{x},y))], \quad (1)$$

where $G$ is the generator, $D$ is the discriminator, $z$ is the latent input, $P_{real}$ is the joint probability distribution of the real data points $(x,y)$, $\tilde{x} = G(z,y)$ refer to the fake samples which are generated by the generator $G$, $E_{x,y \sim P_{real}}[f(.)]$ is the expected value of the function $f(.)$ when the pair $(x,y)$ is drawn from $P_{real}$ and $E_{\tilde{x} \sim G(z,y)}[f(.)]$ is the expected value of the function $f(.)$ when the synthetic sample $\tilde{x}$ is drawn from $G$.

To further evaluate the impact of different generative approaches on data fidelity, we explored two additional generators: (i) the Tabular Variational Autoencoder (TVAE) [10, 11], and (ii) the Tabular Diffusion Probabilistic Model (TabDDPM) [12]. The TVAE was implemented using the SDV library [10, 11] and it was used to model the latent distribution of tabular data using a multivariate Gaussian prior and reconstruct synthetic samples via decoder networks. A lightweight version of the TabDDPM [12] was implemented. In brief, the TabDDPM progressively corrupts the input data through a forward noise process and learns to reverse this corruption using a denoising network trained over multiple diffusion steps. To better examine the value of synthetic data and to avoid data leakage effects, the CTGAN, TVAE, and TabDDPM were trained on the real train set to produce the minority samples needed to reach a 1:1 ratio in the target class.

To assess how well synthetic samples match the real data samples, we computed the Kullback-Leibler divergence (KLD) over feature marginals:

$$KL(P_{real}(x_i)||P_{synthetic}(x_i)) = \sum_{x_i} P_{real}(x_i) log \frac{P_{real}(x_i)}{P_{synthetic}(x_i)}, \quad (2)$$

where $P_{real}(x_i)$ and $P_{synthetic}(x_i)$ denote the empirical (real) and synthetic probability distribution of a feature $x_i \in x$ (from the input dataset). Low KL values across features indicate good fidelity in reproducing the statistical structure of the original data. In addition to the KLD, we computed: (i) the Jensen-Shannon Divergence (JSD) to measure the similarity between the probability distributions, (ii) the Kolmogorov-Smirnov Distance (KSD) to quantify the maximum difference between the cumulative distribution functions of the real and synthetic data, and (iii) the Average Correlation Difference (ACD) to capture the extent to which feature-to-feature relationships (correlation structure) are preserved in the synthetic data.

For comparison purposes, the random downsampling with replacement strategy was also applied. To this end, we refer to the random downsampling with replacement CIH strategy as CIH1 and to the synthetic data generation-based class imbalance handling (CIH) strategy as CIH2.

### E. AI modeling stage

Two classification tasks were defined, namely: (i) CT1: diagnosis of cardiovascular disease (CVD), and (ii) CT2: comorbid diagnosis of CVD and depression. The Extreme Gradient Boosting (XGBoost) algorithm [13] as a high-performance ensemble method known for its efficiency and predictive accuracy in structured data problems. XGBoost is based on the gradient boosting framework, where an ensemble of weak learners—typically decision trees—is trained sequentially, with each new model focusing on correcting the errors made by its predecessors. For comparison purposes, the Random Forest bagging algorithm was also deployed. Both the XGBoost and the Random Forest (RF) classifiers were trained on the CTGAN-augmented training set and on the randomly downsampled training set (including the optimized subset of features identified through mutual information ranking and elbow-point selection). Hyperparameters were finally tuned using default heuristics to maintain computational efficiency and achieve competitive performance. While downsampling (CIH1 strategy) aimed to reduce the class skew at the data level, we also applied XGBoost's "scale_pos_weight" parameter [12] during training to account for any residual imbalance and reinforce sensitivity to the minority class.

### F. AI model performance evaluation stage

The performance of the trained models was evaluated on the independent test set to provide an unbiased assessment of its generalization capability. Multiple standard classification metrics were computed to capture different aspects of model performance, including accuracy (ACC), sensitivity (SENS), specificity (SPEC), and the area under the receiver operating characteristic curve (AUC). Furthermore, feature importance analysis was conducted using XGBoost's and RF's built-in gain-based importance scores. These scores quantify the contribution of each feature to the model's decision trees by evaluating the improvement in purity (information gain) associated with the splits. These plots highlight key clinical and metabolomic variables that drive the model's predictions.

### III. RESULTS

### A. Class imbalance handling

According to Fig. 2, the generated synthetic data exhibit a strong ability to replicate the empirical distribution of the real data (Outcome CVD: KSD 0.17, JSD: 0.16, ACD: 0.06; Outcome Depression and CVD: KSD 0.28, JSD: 0.04, ACD: 0.1). In variables such as the number of times heard an un-real voice and number of times seen an un-real vision, the synthetic distributions closely follow the multimodal peaks of the real data, preserving both skewness and kurtosis. These features are highly sparse and right-skewed, which suggest that the CTGAN effectively learned rare-event structures which is a crucial attribute for modeling psychiatric comorbidity.

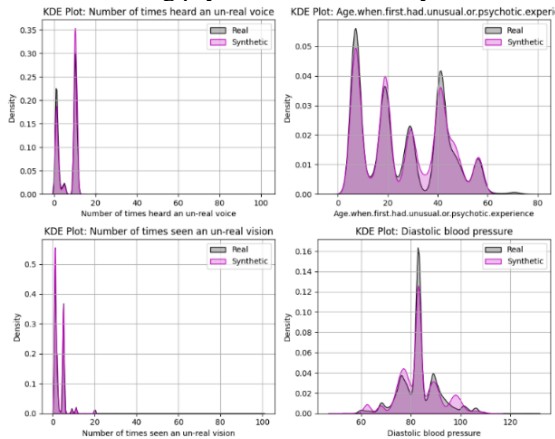

**Fig. 2.** Kernel density estimation plots between synthetic and real data for four randomly selected features.

According to Table I, the CTGAN and TVAE produced synthetic data with lower distributional shifts, as indicated by lower KLD, KSD and JSD, and better preservation of inter-feature relationships, as shown by lower ACD, especially in the case of the CVD outcome. The KLD values remain comparatively low for CTGAN (0.21) and TVAE (0.19), whereas TabDDPM shows a higher KLD (0.28), suggesting less similarity in the marginal distributions of real and synthetic features. For the Depression and CVD combined outcome, a similar pattern is observed: CTGAN maintains moderate fidelity (KLD = 0.24, KSD = 0.15, JSD = 0.23, ACD = 0.23), while TVAE performs well in distributional alignment (JSD = 0.14) but shows a slightly higher KLD (0.32) and ACD (0.28). TabDDPM consistently yielded higher divergence (e.g., KLD = 0.39, KSD = 0.54) and less correlation preservation (ACD = 0.30), indicating that further improvements are needed for it to match the fidelity levels achieved by CTGAN and TVAE.

TABLE I.    SYNTHETIC DATA FIDELITY ASSESSMENT RESULTS PER SYNTHETIC DATA GENERATOR AND PER OUTCOME.

| Metric | CTGAN | TVAE | TabDDPM |
|---|---|---|---|
| **Outcome: CVD** | | | |
| **KLD** | 0,21 | **0,19** | 0,28 |
| **KSD** | **0,18** | 0,22 | 0,35 |
| **JSD** | **0,06** | **0,06** | 0,11 |
| **ACD** | **0,11** | 0,13 | 0,21 |
| **Outcome: Depression and CVD** | | | |
| **KLD** | **0,24** | 0,32 | 0,39 |
| **KSD** | **0,15** | 0,25 | 0,54 |
| **JSD** | 0,23 | 0,14 | **0,23** |
| **ACD** | **0,23** | 0,28 | 0,30 |

### B. AI modeling performance

Table II summarizes the classification results across: (i) the classification tasks CT1 (Outcome: CVD) and CT2 (Outcome: CVD and depression), (ii) the class imbalance handling strategies CIH1 (random downsampling with replacement) and CIH2 (synthetic data generation), and (iii) the two classifiers XGBoost (XGB) and Random Forest (RF). Synthetic data augmentation was applied using the synthetic data of the CTGAN since it yielded synthetic data with the highest overall fidelity (Table I). For CT1, the AI models trained on the synthetically augmented data consistently outperformed those trained on the downsampled data. Specifically, the XGBoost model trained on the synthetic augmented data achieved the highest overall accuracy (0.91), specificity (0.94), and AUC (0.96), indicating enhanced discriminative ability while preserving sensitivity (0.85). While downsampling yielded strong sensitivity (0.89) in the XGBoost model, it yielded lower specificity (0.90) and slightly reduced AUC (0.96). The RF models followed a similar trend, where synthetic data augmentation achieved better specificity and AUC compared to the downsampling strategy.

TABLE II.    AI MODELING PERFORMANCE ANALYSIS RESULTS.

| Task | CIH Strategy | Model | Train Size | ACC | SENS | SPEC | AUC |
|---|---|---|---|---|---|---|---|
| CT1 | CIH1 | XGB | 79664 | 0.90 | 0.89 | 0.90 | 0.96 |
| | | RF | 79664 | 0.85 | 0.84 | 0.87 | 0.93 |
| | CIH2 | **XGB** | **139674** | **0.91** | **0.85** | **0.94** | **0.96** |
| | | RF | 139674 | 0.86 | 0.82 | 0.89 | 0.93 |
| CT2 | CIH1 | XGB | 4522 | 0.77 | 0.86 | 0.76 | 0.89 |
| | | RF | 4522 | 0.68 | 0.91 | 0.65 | 0.86 |
| | CIH2 | **XGB** | **47590** | **0.87** | **0.76** | **0.88** | **0.92** |
| | | RF | 47590 | 0.81 | 0.71 | 0.82 | 0.84 |

The comorbid classification task (CT2) posed a greater challenge due to its smaller training set size and heterogeneous phenotype patterns. Downsampling (CIH1) led to the highest sensitivity (0.91) in the RF model but with reduced specificity (0.65) and lower accuracy (0.68). In contrast, synthetic data augmentation (CIH2) yielded more balanced results across metrics. The XGBoost model trained on synthetic data achieved the highest accuracy (0.87) and specificity (0.88) (Fig. 3). Notably, this model also reached the highest AUC (0.92).

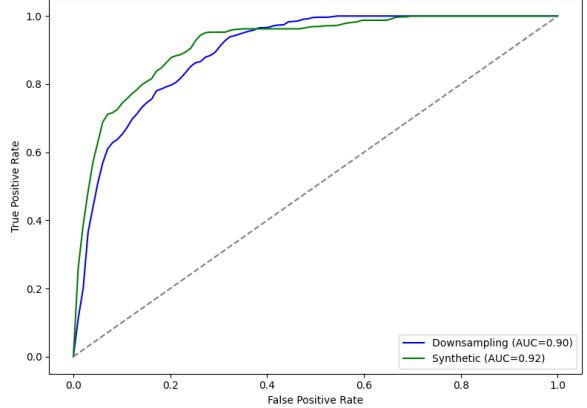

**Fig. 3.** Comparison of the ROC curves (XGBoost) between the class imbalance handling strategies CIH1 (blue: downsampling) and CIH2 (green: synthetic augmentation) for the classification task CT2 (CVD and depression).

### IV. DISCUSSION

In this study, we developed an AI-driven pipeline that integrates clinical and NMR-based metabolomics data from the UK Biobank to support the diagnosis of cardiovascular disease (CVD) and its comorbidity with depression. A key focus was the systematic comparison between two class imbalance handling strategies: random downsampling with replacement (CIH1) and synthetic data augmentation using CTGAN (CIH2). The findings of the current study consistently demonstrate the superiority of synthetic augmentation in achieving balanced model performance, particularly for complex comorbid classification tasks.

As shown in Table II, for the primary task of diagnosing CVD (CT1), the XGBoost model trained on synthetically augmented data (CIH2) achieved the highest accuracy (0.91), specificity (0.94), and AUC (0.96), while maintaining a sensitivity of 0.85. In contrast, the downsampling approach (CIH1) led to slightly lower specificity (0.90) and accuracy (0.90), despite yielding a higher sensitivity (0.89). These results suggest that while downsampling may enhance sensitivity, it may compromise specificity and overall model generalization which is an important trade-off in clinical applications where false positives can lead to unnecessary interventions. The comorbid classification task (CT2) presented more significant challenges due to a smaller training size (4,522 for CIH1 against 47,590 for CIH2) and more heterogeneous phenotype distributions. While CIH1 achieved sensitivity 0.91 in the RF model, its specificity dropped to 0.65, and accuracy to 0.68. On the other hand, the XGBoost model trained on synthetic data attained a higher accuracy (0.87), specificity (0.88), and

sensitivity (0.76), resulting in the highest AUC (0.91) across all models (Table II). These outcomes support the utility of CTGAN in capturing complex multimodal dependencies and enhancing classifier robustness under data imbalance.

Finally, in terms of data fidelity, the synthetic data closely matched the distribution of the original dataset (Fig. 2), where key features such as the number of times participants reported hearing voices or experiencing psychotic symptoms show strong alignment between synthetic and real distributions. CTGAN and TVAE outperformed TabDDPM in terms of preserving data fidelity across several statistical measures (Table I). Specifically, for the CVD outcome, CTGAN achieved the lowest KLD (0.21), JSD (0.06), and ACD (0.11), indicating minimal distributional shift and better structural preservation. TVAE showed similarly competitive performance, with slightly lower KLD (0.19) but marginally higher ACD (0.13). In contrast, TabDDPM yielded higher divergence values (KLD = 0.28, JSD = 0.11, ACD = 0.21), suggesting reduced fidelity in capturing real data patterns. The trend was consistent in the Depression and CVD case, with TabDDPM again showing the highest divergence (KLD = 0.39, KSD = 0.54). These findings underline the importance of selecting an appropriate generator tailored to the characteristics of tabular biomedical data when aiming for high-fidelity synthetic augmentation.

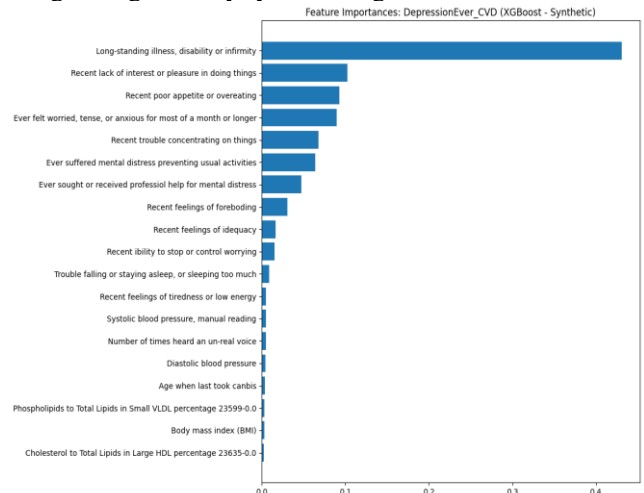

**Fig. 4.** Feuture importance for the diagnosis of ever having depression and CVD using synthetic data and the XGBoost classifier.

From the clinical point of view, our work presents a novel strategy for the diagnosis of depression and CVD comorbidity. As shown in Fig. 4, the features of the developed model include several behavioral and social questions, but also some related to CVD prevalence. The biomarker of phospholipids in very low density lipoproteins is associated with the presence of CVD and it is in agreement with the current literature [14]. As feature work, we plan to investigate the role of all the NMR based metabolomics biomarkers found in the developed models and associate them with the prevalence of multimorbidity conditions in CVD, mental and metabolic disorders. A limitation of the current study is the reliance on retrospective data, which may not capture real-time clinical variability or emerging risk factors. Overall, our findings highlight the value of synthetic data augmentation to: (i) enhance diagnostic models, especially in underrepresented clinical scenarios like CVD and depression comorbidity, (ii) support earlier and more

accurate diagnosis, reducing the burden of missed or misclassified cases.

In terms of translational insights, several of the top-ranked NMR biomarkers which have been identified in our XGBoost feature importance analysis, such as phospholipids in very-low-density lipoproteins (VLDL), are mechanistically linked to both cardiovascular risk and depression [15]. For instance, VLDL particles have been implicated in endothelial dysfunction and systemic inflammation, which are shared pathophysiological pathways in CVD and major depressive disorder [16]. Other selected metabolites (e.g., glycoprotein acetyls, ketone bodies) are also known to reflect chronic low-grade inflammation, oxidative stress, and energy metabolism dysregulation, all of which are relevant in the biopsychosocial model of CVD–depression comorbidity.

## V. CONCLUSIONS

In this work, advanced synthetic augmentation (via CTGAN) was combined with class-imbalance mitigation to build robust AI models for the cardiovascular–depression comorbidity. The proposed hybrid workflow integrates downsampling, "scale_pos_weight" adjustments in the XGBoost, and GAN-driven oversampling to overcome the limitations of conventional class imbalance handling methods. The resulting AI model was able to achieve an AUC 0.96 while preserving complex, multimodal clinical and metabolomic feature distributions. From a clinical point of view, key NMR biomarkers were identified, including VLDL phospholipids, glycoprotein acetyls, and ketone bodies that shed light on shared inflammatory and metabolic pathways. These findings can support early patient stratification and personalized interventions for those at risk of both CVD and major depressive disorder. The current work promotes the use of generative modeling in healthcare and highlights its potential to enhance diagnostic accuracy and improve patient outcomes by supporting unbiased internal validation and a clear roadmap for external cohort testing.

As part of our future work, we plan to expand this mechanistic interpretation in follow-up studies using pathway enrichment and mediation analysis to better connect the metabolic signatures with clinical outcomes. In addition, we aim to apply the proposed workflow to other biobanks or cohort studies (e.g., TwinsUK [17], Lifelines [18]) that include comparable omics and clinical variables. Finally, we plan to investigate the integration of weighted loss functions with synthetic data augmentation to assess whether such hybrid approaches could yield additional gains in robustness and minority class sensitivity.

## VI. ACKNOWLEDGEMENT

This project has received funding from the European Union's Horizon 2020 research and innovation programme TO_AITION under grant agreement No 848146.

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
