# OpenReview forum: "Enhancing the diagnosis of CVD and depression comorbidity through the augmentation of synthetic metabolomics data"
_IEEE.org/EMBS/BHI/2025/Conference — BHI 2025_

### Official Review · Reviewer_NWip · 2025-06-25
**diagnosing cardiovascular disease (CVD) and its comorbidity with depression**

**Confidence:** 4
**Clarity Of Writing:** good
**Clinical Significance:** great
**Methodological Novelty:** good
**Overall Rating:** 6

**Experiments And Results:**

good

**Questions For The Authors:**

Have the authors considered evaluating their approach using other generative models (e.g., VAEs, diffusion models) to benchmark CTGAN's performance?
Could the authors provide more insight into how the selected NMR biomarkers relate mechanistically to CVD–depression comorbidity? This could enhance translational significance.

**Strengths:**

The integration of NMR metabolomics adds a rich layer of biological insight often overlooked in purely clinical ML studies.
Strong evaluation framework across multiple metrics (accuracy, sensitivity, specificity, AUC), with clear evidence favoring synthetic augmentation.

**Summary Of The Paper:**

This paper presents a synthetic data augmentation approach using CTGAN to improve classification performance in diagnosing cardiovascular disease (CVD) and its comorbidity with depression. Leveraging multimodal data (clinical and NMR-based metabolomics) from the UK Biobank, the authors compare synthetic oversampling with traditional random downsampling across two tasks: CVD diagnosis and comorbid CVD–depression classification. AI models trained on synthetically augmented data demonstrate superior performance, especially in terms of specificity and AUC. Fidelity between real and generated data is validated through KL divergence, JSD, and average correlation difference.

**Weaknesses:**

Feature was mistakenly input as "Feuture" in figure 4 legend.
study is based on one data source: based solely on UK Biobank data; external validation or simulation of real-world deployment conditions is lacking.

---

### Official Review · Reviewer_jhfo · 2025-07-14
**Synthetic Data Helps CVD–Depression Prediction**

**Confidence:** 3
**Clarity Of Writing:** excellent
**Clinical Significance:** great
**Methodological Novelty:** good
**Overall Rating:** 5
**Final Rating:** 6

**Experiments And Results:**

good

**Questions For The Authors:**

Could you clarify whether you tested the synthetic augmentation approach on any external or independent datasets?

**Strengths:**

The paper addresses a clear and important challenge in medical AI — class imbalance in comorbidity prediction — by applying a modern synthetic data approach.
Using CTGAN for tabular clinical and metabolomics data is well motivated and goes beyond simple resampling. The results show consistent improvements in diagnostic performance, especially for the difficult CVD and depression comorbidity task. The authors also provide fidelity metrics to verify that the synthetic data preserves the original data structure. Combining large-scale, multi-modal data (clinical plus NMR metabolomics) and interpretable feature importance adds practical value for future clinical research and potential real-world application.

**Summary Of The Paper:**

This paper presents an AI-based pipeline to improve the diagnosis of cardiovascular disease (CVD) and its comorbidity with depression by handling severe class imbalance in clinical and metabolomics data from the UK Biobank. The authors compare traditional random downsampling with synthetic data augmentation using a Conditional Tabular GAN (CTGAN). They show that models trained with synthetic data achieve better performance (e.g., higher accuracy and AUC) for both CVD and comorbid CVD–depression prediction tasks. The fidelity of the synthetic data to real data is evaluated with metrics like KL divergence, JSD, and ACD, showing good alignment. Feature importance analysis is used to interpret model predictions. The work demonstrates that synthetic data augmentation can improve diagnostic performance in imbalanced, high-dimensional biomedical datasets.

**Weaknesses:**

While the study shows promising gains with synthetic data, it relies only on retrospective UK Biobank data, so the real-world clinical benefit remains untested.
The paper does not explore how well the synthetic data generalizes across external cohorts or whether it risks introducing subtle biases in practice
. Additional experiments using independent validation sets or prospective data would strengthen the evidence that CTGAN-based augmentation works reliably. The paper could also benefit from more discussion on how to translate these models into clinical workflows, including how synthetic samples interact with privacy, regulatory, and explainability concerns in sensitive health settings.

---

### Official Review · Reviewer_Erp8 · 2025-07-17
**Enhancing the diagnosis of CVD and depression comorbidity through the augmentation of synthetic metabolomics data**

**Confidence:** 3
**Clarity Of Writing:** good
**Clinical Significance:** fair
**Methodological Novelty:** fair
**Overall Rating:** 6

**Experiments And Results:**

good

**Questions For The Authors:**

· Have you considered exploring the use of a weighted loss function to address the data imbalance, and if so, what were the outcomes or rationale for your chosen approach?
· Have you investigated other techniques for enhancing performance?

**Strengths:**

The work is presented with clarity, offering a good comparative analysis of integrating synthetic data across several models. Furthermore, the inclusion of well-designed diagrams significantly enhances the comprehensibility of the methodology and findings, making complex concepts readily accessible to the reader.

**Summary Of The Paper:**

This paper introduces an AI-driven pipeline that leverages synthetic data augmentation to enhance the diagnosis of cardiovascular disease and depression, addressing challenges posed by complex phenotypes and severe class imbalance in real-world health data. The study, uses clinical and NMR-based metabolomics data and compares traditional random downsampling with synthetic data augmentation using CTGAN.

**Weaknesses:**

To further strengthen the study, it would be beneficial to include an analysis of the real data distribution, providing deeper insights into its characteristics. Additionally, evaluating the model's performance on a different, non-local test partition would also be essential to assess its inter-database generalization capabilities and confirm its robustness across datasets.

On a separate note, there appears to be a formatting inconsistency on the last page. Specifically, below Figure 4, there's a mismatch in letter size. This text seems to describe future work rather than serving as a figure caption, and it should be reviewed and corrected for clarity and proper presentation.